# Fodder Galega—Persistence as a Special Asset in Sustainable Agriculture

Stanisław Ignaczak, Jadwiga Andrzejewska * and Katarzyna Sadowska

Agronomy Department, Faculty of Agriculture and Biotechnology, Bydgoszcz University of Science and Technology, 85-796 Bydgoszcz, Poland; staszekmig@o2.pl (S.I.); katarzyna.sadowska@pbs.edu.pl (K.S.)
* Correspondence: jadwiga.andrzejewska@pbs.edu.pl

**Abstract:** Perennial crops, especially legumes, have a crucial role in the development of sustainable agriculture. One such species may be fodder galega (*Galega orientalis* Lam.), whose utility values, including persistence and multi-directional use, are still not sufficiently known and appreciated. The aim of this study was to evaluate fodder galega yield dynamics, taking into account some indices of fodder value and the accumulation of mineral components in long-term use in light soil under moderate climate conditions with periodic shortages of precipitation. The results of six long-term experiments conducted at the Mochełek Research Station (53°120′ N, 17°510′ E) were evaluated. The dynamics of dry matter, total protein yield, and accumulation of minerals were best reflected by trend lines in the form of logarithmic functions, where during the first 4–5 years of use, a significant increase in the assessed values was noted, and in subsequent years, the increase continued but at a lower rate. The advancement in plant development in the establishment year had a significant impact on yields in the first production year. On average, during 10 production years, the dry matter yield obtained was 936 g m$^2$, and the total protein yield was 177 g m$^2$. Between 50% and 60% of the yield was determined by herbage accumulation in the first cut. Among minerals, the highest accumulation level was achieved for potassium. The height of shoots and the content of crude fiber in the plants increased linearly in the following years. The shoot density, leafiness, and content of minerals did not depend on the age of the galega stand, and their values were different among growth periods within a season. The number of shoots per 1 m$^2$ in successive cuts was 170, 139, and 92, and their height was 79, 67, and 31 cm, respectively. The share of leaves in the first cut yield was 50%, and in the second and third cuts, it was 65% on average. In these conditions, over 10 years of use, galega turned out to be a valuable, persistent, and reliably yielding fodder crop.

**Keywords:** *Galega orientalis*; long-term use; yield dynamics; accumulation of macroelements; shoots density; legume; fodder

## 1. Introduction

Intensively managed, high-yielding annual crops play a dominant role in modern agriculture. Frequent tillage and a lack of vegetation cover for prolonged periods typical of these farming systems have led to extensive soil erosion, soil carbon loss, and nutrient runoff into surface water, which have a destructive effect on agroecosystems. This problem is growing with climate change, especially rising temperatures, intense rainfall events, as well as permanent or periodic water deficits. Under these circumstances, perennial crops gain in importance. The shift toward perennial crops is a global trend and applies not only to fodder crops, but also to grain and seed crops [1,2]. This approach is in line with the idea and framework of sustainable agriculture [3]. Since perennial crops last beyond one season, the disturbance needed for establishment can be amortized over multiple production years. Therefore, species and cultivars of perennial crops should be well adapted to difficult environmental conditions, be high-yielding with the possibility of multi-directional use or crop management, and be resistant to pathogens. These criteria are met by several species

from the group of perennial legumes, which additionally do not require costly nitrogen fertilizers and, moreover, enrich the soil with nitrogen and organic matter. One of the species with the longest utilization period described in the scientific literature is fodder galega (*Galega orientalis* Lam.) [4–6].

Galega originated from the foothills of the Caucasus. In Europe, this legume was first investigated as a forage crop in Finland and Estonia [7,8]. Currently, galega is cultivated in Finland, Latvia, Estonia, and Russia, but results of research published in over 25 years indicate that the range of its cultivation could be extended to other European countries, including Sweden, Denmark, Poland, Moldova, and Ukraine [9–12]. Recent research also demonstrates that galega is also adapted to some regions of Canada and Japan [13,14].

Initially, galega was treated as a fodder crop intended for pasture or mechanical harvest [8,15]. Currently, it is perceived as a more versatile crop because, in addition to its importance as a source of fodder, its potential for the protection of fallow lands and remediation of contaminated soils has also been documented [16,17]. Moreover, it is recommended as a productive and sustainable energy crop both for direct combustion and for biogas production [18]. It is also possible to use the leaves for fodder and the stems as organic mulch to cover the soil [19].

The versatility of galega as a crop is a consequence of its biological properties. The root system is a taproot type, with a large number of lateral roots that penetrate the soil to a depth of 60–70 cm. On the underground part of the stem, 3–4 overwintering buds are formed annually, and rhizomes are formed on the root collar, which, after emerging from the soil, turn into stems. Buds at the base of upright stems and on rhizomes enable the above-ground parts of the plant to regrow in spring and after each harvest. The stems of eastern galega are quite stiff, erect, and usually reach a height of 80 to 130 cm. The leaves are odd-pinnately compound and 15–30 cm long. Blue-violet flowers are an attractive nectar source for bees and other pollinators. The fruit is a 3 cm long, indehiscent pod usually containing 3 to 7 seeds. The thousand seed weight ranges from 5 to 9 g. Most galega seeds are hard by nature and require scarification before sowing [20,21].

One of the most difficult elements of galega cultivation technology is establishing and maintaining the stand in the first year. The basic conditions for success are the scarification of the seed and inoculation with the appropriate strain of *Rhizobium galegae* [22,23]. Plant development in the first year is very slow; therefore, the crop is exposed to weed infestation and competition. Studies have rarely provided information on the management and yield of galega in the establishment year.

The principal advantage of galega, compared to other legumes, is its persistence, but the majority of scientific studies have reported results of single experiments of a 3 to 6-year duration. There have been few studies on longer periods of utilization and, as a rule, the level of yields is given without an analysis of environmental conditions and management practices [4–6,24]. It has also been also reported that in the 5–7th production years, there is a significant decrease in yields [4,25]. One of the important factors determining the yield level is the number of harvests. The highest yields perform by collecting 2 or 3 cuts during the growing season [9].

Despite the significant positive attributes of this species, its adoption as an agricultural crop has been very slow. With current climate change trends and the EU's agricultural policy aimed at supporting biodiversity, galega should find a prominent place among crops. However, the greater promotion of this species is warranted based on the results of field experiments conducted especially in light soils, where there are periodic shortages of rainfall. These criteria correspond to the results presented in this paper. The experiments were carried out in the 1980s and 1990s, but due to the level of yields reported in more recent publications being comparable to earlier publications [4,25,26], the results reported here remain valid. Their value is also determined by the fact that they are a synthesis of data from several field experiments, and they also address insufficiently known issues, including the yield structure and the dynamics of the accumulation of dry matter and nutrients in the establishment year and long-term use.

The aim of the study was to evaluate fodder galega yield dynamics, taking into account indices such as fodder value and accumulation of minerals, in long-term use in light soil under moderate climate conditions with periodic shortages of rainfall.

## 2. Materials and Methods

### 2.1. Experimental Site and Field Operations

The subject of this synthesis of fodder galega characteristics is a monoculture crop used for green fodder in several experiments conducted at the Mochełek Research Station (53°120 N, 17°510 E), Poland, where the soil is a Cutanic, Haplic Luvisol. Galega was grown in soils that had a low C content and a C:N ratio that was narrow but typical of legume fields. The content of assimilable forms of nutrients was as follows: phosphorus—medium or high, potassium—low or medium, magnesium—medium. The pH of the soils ranged from acidic to neutral, usually slightly acidic (Table 1). The thermal conditions were typical of the temperate climate, but periods with very low rainfall were common, especially during the spring growth cycle (Table 2).

**Table 1.** Characteristics of soil conditions at the Mochełek Research Station (means from 1982 to 1999).

| Value | C [%] | N [%] | C:N | Assimilable Forms [mg 100 g] | | | $pH_{(KCl)}$ |
| --- | --- | --- | --- | --- | --- | --- | --- |
| | | | | $P_2O_5$ | $K_2O$ | MgO | |
| min | 0.63 | 0.041 | 13.3 | 14.0 | 6.0 | 1.5 | 5.0 |
| max | 1.22 | 0.092 | 15.4 | 22.3 | 13.0 | 4.4 | 6.7 |
| mean | 0.78 | 0.051 | 14.9 | 17.9 | 10.9 | 3.3 | 5.6 |
| most common | 0.63 | 0.041 | 15.4 | 17.2 | 13.0 | 4.1 | 6.7 |

**Table 2.** The length of the growing period and weather conditions during successive growth cycles at the Mochełek Research Station (means from 1982 to 1999).

| Parameter | Growth Cycle | Value | | |
| --- | --- | --- | --- | --- |
| | | Maximum | Minimum | Mean |
| | I | 90 | 31 | 57 |
| Number of days of growth | II | 97 | 30 | 54 |
| | III | 92 | 52 | 69 |
| | I | - | - | 10.6 |
| Mean daily air temperature (°C) | II | - | - | 17.1 |
| | III | - | - | 15.5 |
| | I | 141 | 12 | 66 |
| Rainfall [mm] | II | 192 | 46 | 104 |
| | III | 225 | 54 | 127 |

In these experiments, we studied the Gale cultivar. The experiments were carried out by sowing galega in pure stands using the seeding rate of 12 kg ha$^{-1}$ or as the under-sown crop in oats for green fodder, and then using 15 kg ha$^{-1}$. The seeds were mechanically scarified and inoculated with *Rhizobium galegae*. Sowing was carried out between 30 April and 17 May, i.e., between the 120th and 137th days of the year. Weeds were controlled mechanically in the establishment year. In the following years, weeds were chemically controlled in early spring before the emergence of galega. Mineral fertilizer was applied at the following rates (kg ha$^{-1}$): in the establishment year and in the first production year, 50 P and 80 K, and in the following production years 100 P every other year and 120 K

annually. Before the establishment of the experiments and after 6 years, the soil was limed with magnesium lime.

The experiments reported here addressed various issues, including: comparison of fodder galega with other fodder plants (alfalfa, clover, and sainfoin, and their mixtures with galega); research on sowing, harvesting, and conservation methods; and research on root system development. Depending on the experiment, the plot area ranged from 1 to 16 m$^2$. The results of most of these experiments have not been previously published.

Six field experiments were established between 1981 and 1993, and the results of full-use years are from 1982 to 1999. Five experiments were used for 6 years. One experiment, established in 1988, was fully utilized and evaluated for 10 years, and then the cultivation was extended by using the crops for cattle feed, and the field with galega was plowed in 2018. However, plant samples were taken from this crop in 2016 and 2017, and the results were used in other publications [19,27]. Galega was mostly harvested three times during the growing season. The first cut was harvested when the plants were at the stage from budding to flowering and had developed 5–8 leaves. The second and third cuts were usually harvested at the shoot elongation stage, when the plants had 4–5 leaves. All measurements were made in four replicates. Due to the differences in the plant harvest area over the years, the yield results were calculated per 1 m$^2$. Only the uptake of elements was given in kg ha$^{-1}$ so that it would be easier to discuss them in relation to other authors' works.

### 2.2. Chemical Analysis of Soil and Plant Materials

Soil fertility analyses were performed using the following methods: C—Tiurin, N—Kjeldahl, P and K—Egner–Riehm, and Mg—Schachtschabel. The content of macroelements in the plant material was determined as follows: total nitrogen—by the Kjeldahl method, crude fiber—by the modified Henneberg and Stohman method, K and Ca—photometrically, P—by the vanadium–molybdenum method, and Mg by the titanium yellow method. The total protein content was calculated as %N $\times$ 6.25.

### 2.3. Statistical Analysis

The results of measurements in specific cuts and in total over the years were presented primarily as functions of plant age in the form of logarithmic or linear functions, depending on the value of the determination coefficient ($R^2$). In the event that the $R^2$ was below 0.5 for the data from all cuts, which indicated an unsatisfactory or poor fit of the data to the regression, the results were given as means with standard deviation. Correlation coefficients were calculated for selected characteristics, and the Guilford classification was used to evaluate them, according to which *r* values above 0.5 are considered high correlation. A Microsoft Excel 2019 (Microsoft Corp., Redmond, WA, USA) spreadsheet was used for the calculations.

## 3. Results

### 3.1. Establishment Year

Plant emergence occurred 11–17 days after sowing. The date of plant harvesting was determined by the development stage or weather conditions indicating the end of the growing period. One cut was harvested in the establishment year. In various years, the harvest took place after 61–125 days of growth, during which the plants were then at various developmental stages, i.e., from the beginning of stem formation to the full flowering stage. In the establishment year, the plants formed only one branching shoot with a height of 43.4 to 103.0 cm. In the first production year, the proportion of leaves in the dry matter yield averaged 42.8% and was negatively and highly correlated with the dry matter yield (Table 3).

**Table 3.** Growth conditions and indices of fodder galega development as well as the coefficient of variation and correlations with dry matter yields in the establishment year.

| Parameter | Value | | | Coefficient of Variation (%) | Coefficient of Variation with DM Yield |
| --- | --- | --- | --- | --- | --- |
| | Minimum | Maximum | Mean | | |
| Dry matter yield (g m$^{-2}$) | 21.5 | 647.2 | 245.5 | 95.7 | 1.000 |
| Fresh matter yield (g m$^{-2}$) | 76.5 | 995.0 | 402.5 | 71.1 | 0.995 |
| Dry matter content (%) | 21.8 | 28.3 | 25.5 | 8.3 | −0.127 |
| Height of shoots (cm) | 43.4 | 103.0 | 78.7 | 31.8 | 0.985 |
| Proportion of leaves in the dry matter yield (%) | 37.0 | 46.4 | 42.8 | 9.8 | −0.868 |

It should be mentioned here that the Gale cultivar was registered in 1988 as the first cultivated cultivar of *G. orientalis* [28]. At the end of the last and the beginning of this century, breeding was carried out to develop new cultivars in Lithuania [29], in Russia [30], and in Poland [31]. The breeding focus was primarily aimed at developing cultivars with high and stable yields of fodder and seeds. However, our results indicate that cultivars with a faster development rate and greater vigor in the year of sowing are also needed. This will largely determine farmers' interest in growing this crop.

*3.2. Production Years*

For the data from three experiments, correlation coefficients were calculated between dry matter yields in the establishment year and yields in the first production year, for the first two cuts (Table 4). The results indicate that the condition of galega plants and the resulting yield level in the establishment year had a significant impact on the yield of the first cut and the total yield in the first production year. The condition of the plants in the establishment year had no effect on the yield of the second cut which was probably mainly shaped by weather conditions.

**Table 4.** Correlation between galega dry matter yields in the establishment year and yields in the first production year.

| Parameter | Establishment Year | First Production Year | | |
| --- | --- | --- | --- | --- |
| | | 1st Cut | 2nd Cut | Total |
| DM yield (g m$^2$) | 264.3 | 253.3 | 187.3 | 426.5 |
| Correlation coefficient | 1.000 | 0.954 | 0.242 | 0.786 |

In our study, the total-season yields of galega dry matter ranged from 689 to 1018 g m$^2$, with a mean value of 954 g m$^2$. Therefore, the yield level did not differ from the range reported by other authors, where galega was used for a much shorter period of time [9,25,26,32]. The yield dynamics of galega plants in the 10-year period of use is best reflected by the logarithmic function indicating a significant increase in yield between the 1st and 5th production years and a further, but slower, increase in the subsequent production years (Figure 1). A similar relationship was seen in the first cut, which had a decisive impact on the total-season yield. Its contribution to the total-season yield ranged from 40 to 68%. The yield of the second cut was not determined by the age of the plantation. However, there was a linear decrease in yields in the third cut. It is worth noting, however, that in the same field in 2016, the yield of the first, second, and third cuts of galega was 317, 349, and 385 g m$^2$, respectively, and in 2017, only two cuts were harvested—161 and 191 g m$^2$ [27]. We cite these later data to indicate that the breakdown in yield trends shown in Figure 1 occurred after 10 years of galega production.

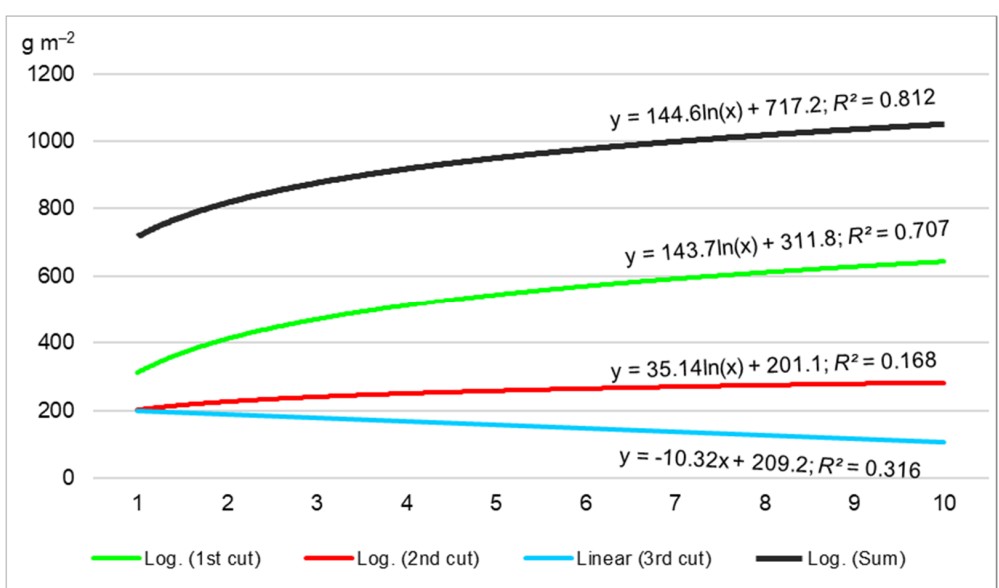

**Figure 1.** Fodder galega dry matter yield as a function of production year.

Zav'yalova et al. [6] reported that the yields of green matter of galega in the 12th and 24th production years were only 9.90 and 10.05 Mg ha$^{-1}$, respectively, with a mean dry matter content of 23.5%. Slepetys and Slepetiene [24] evaluated the yield of galega in an organic farm over 9 production years, where the mean yield was 3.58 Mg ha$^{-1}$ with a coefficient of variation of 30% and without a clearly marked downward trend. Dubis et al. [4] provided the results of yields in an 11-year cultivation period, where the annual dry matter yields increased systematically from the 1st to the 5th years (from approx. 4 to 15 Mg ha$^{-1}$) and then decreased to the level of 5 Mg ha$^{-1}$ in the last year. Moller et al. [9] reported an almost 50% decrease in dry matter yields progressing between the second and seventh years after establishment. However, Adamovics et al. [5] reported that in Latvia, even after more than 20 years of cultivation, galega yields remained at the level of approximately 9 Mg ha$^{-1}$.

The comparison of the present study results and those reported in the literature shows that there is no rule regarding the galega yield dynamics. In the present study, the number of shoots per 1 m$^2$ determined the all-year yield, especially the yield of the third cut. The yield was also significantly correlated with plant height (Table 5). Therefore, the decrease in yield in other experiments was also likely related to these two characteristics.

**Table 5.** Correlation relationships between fodder galega yields and yield structure components.

| Dry Matter Yield | Number of Shoots per 1 m$^2$ | Height of Shoots | Proportion of Leaves in Yield |
|---|---|---|---|
| All-year: 1.00 | 0.687 | 0.885 | −0.803 |
| 1st cut: 1.00 | 0.338 | 0.663 | −0.542 |
| 2nd cut: 1.00 | 0.243 | 0.714 | −0.040 |
| 3rd cut: 1.00 | 0.845 | 0.477 | −0.222 |

The mean number of shoots before harvesting the successive cuts was 170, 139, and 92 with the standard deviation of 45.9, 27.5, and 41.7, respectively (Figure 2). The number of shoots was very diversified in the years of the study, especially in the first cut. This proves that the weather conditions in early spring, which determine the growth of shoots from overwintering buds on the underground part of the stem and from rhizomes, have a significant impact on this characteristic. Relatively large differences between the years were also noted in the third cut. Beginning with the fourth production year, the number of

shoots per 1 m² was always lower than in the first two cuts, with the only exception being the seventh production year.

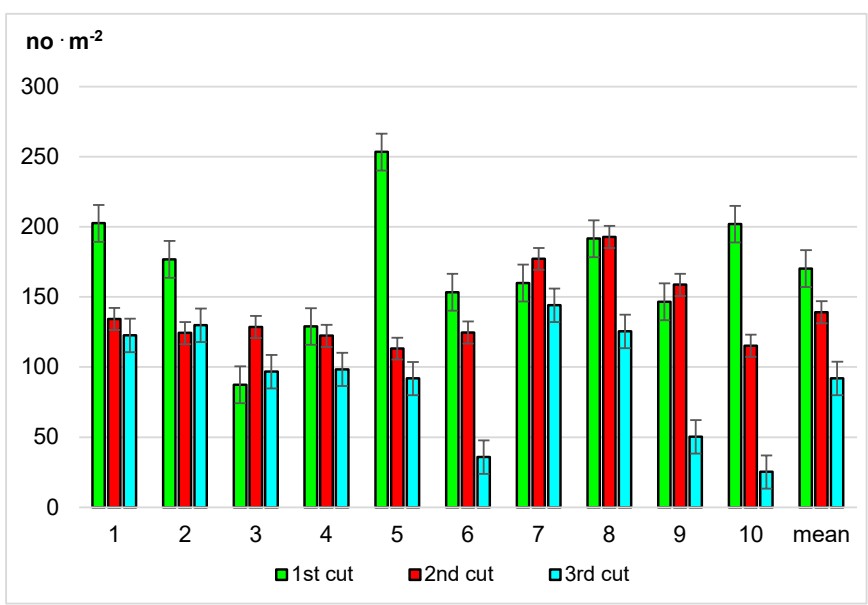

**Figure 2.** Number of galega shoots per 1 m² in cuts and production year.

It was found that in the subsequent production years, the plants in the first and second cuts produced increasingly taller shoots, which is reflected in logarithmic and linear regressions with the $R^2$ coefficients of 0.57 and 0.53, respectively (Figure 3). This could be related to the progressive development of the root, which also enabled the development of the above-ground parts. However, the tendency to grow longer stems must have changed after the 10th production year because in 2016 and 2017, the height of the plants from the first cut was 90 and 58 cm, respectively, and from the second, 114 and 67 cm [27].

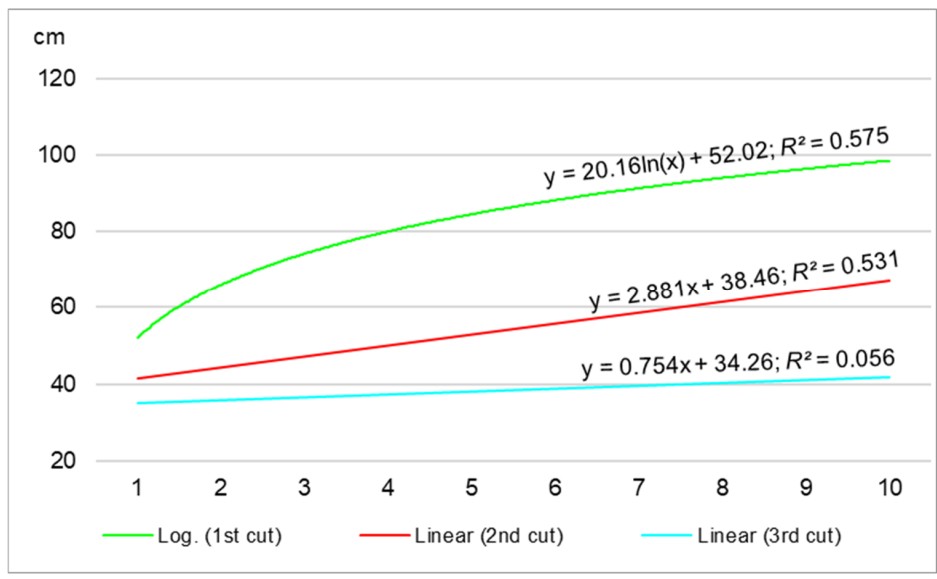

**Figure 3.** Galega shoot height in cuts as a function of production year.

The mean proportion of leaves in the yield of the first cut was 50.2, the second 64.5, and the third 66.0%, with the standard deviation being 3.56, 2.63, and 6.99, respectively (Figure 4). The foliage of the plants was not related to the stand age, which is also evidenced by the results from 2016 and 2017 when the proportion of leaves in the yield of the first cut

was even higher than in the first ten production years and amounted to 58.2 and 62.2%, respectively [27].

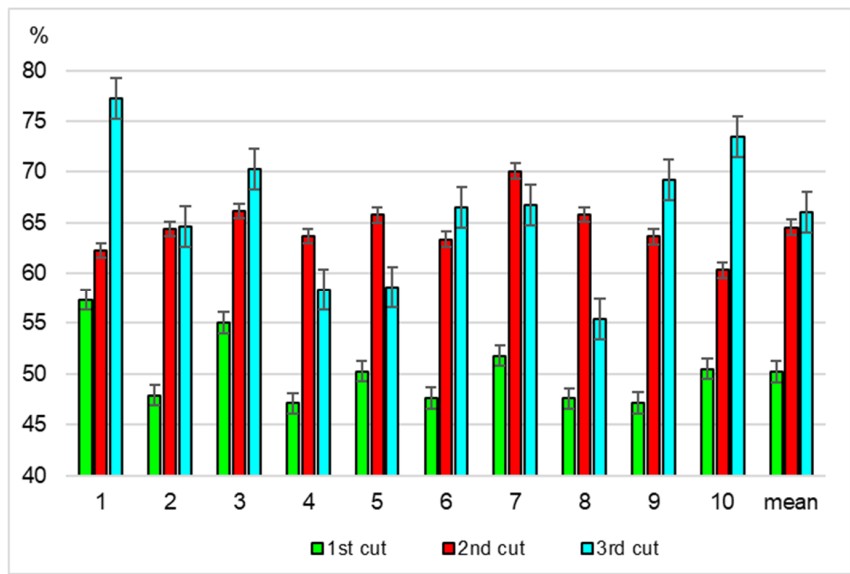

**Figure 4.** The proportion of leaves in galega dry matter yield in production years and in cuts.

The mean fiber content in successive cuts was 304, 289, and 277 g kg$^{-1}$. It was shown that the crude fiber content increased with the age of the galega stand, with the results best fitting the linear function obtained for the data from the second cut (Figure 5). The level of fiber content in plants from the first and third cuts was much more diverse, as evidenced by the relatively low $R^2$ coefficient. However, these are not isolated results because Møller and Hostrup [33] provide data showing how much (within one experiment) the fiber content can increase in a few days, especially in the first but also in the third cut. However, it should be noted that in the present study, in the following years, the plants were getting taller (Figure 3), and this might have resulted in an increase in their crude fiber content.

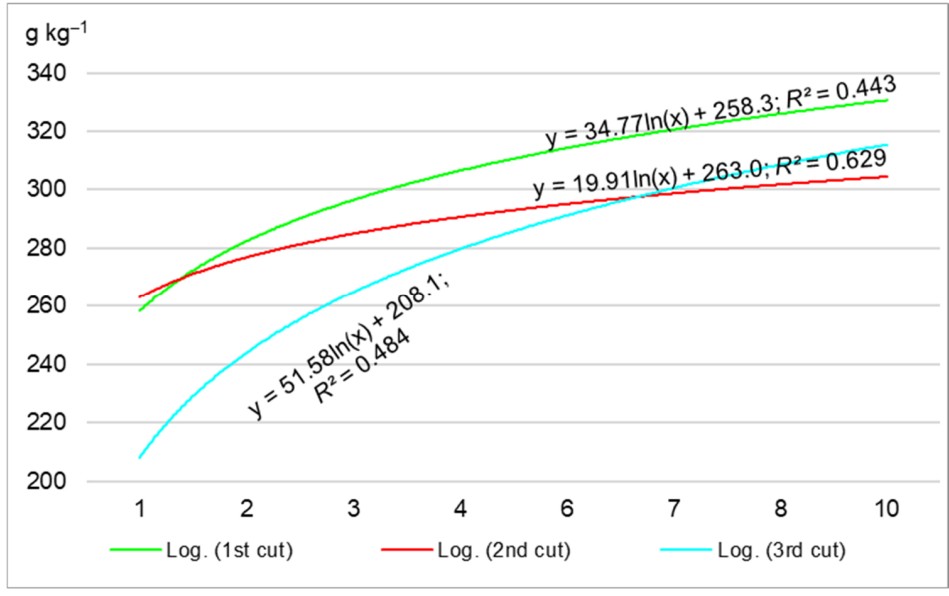

**Figure 5.** Crude fiber content in galega plants in cuts as a function of production year (not determined in the 5th and 9th production years).

The content of individual macronutrients in galega plants in individual cuts was quite variable; therefore, the minimum, maximum, and mean values as well as the production years to which these values refer are presented (Table 6). Noteworthy is the relatively high content of potassium compared to the results reported by Symanowicz and Kalembasa [25] and Baležentienė and Spruogis [34] and the tendency to decrease the content of N, P, and K in successive cuts. Symanowicz and Kalembasa [25] also reported that the first and third cuts have the highest content of the above-mentioned elements, and the second cut has the lowest.

**Table 6.** Content of macroelements (g kg$^{-1}$) in galega dry matter yields from individual cuts.

| Value | I Cut | | II Cut | | III Cut | |
|---|---|---|---|---|---|---|
| | Content | Production Year | Content | Production Year | Content | Production Year |
| Total nitrogen | | | | | | |
| min | 25.7 | 5 | 24.7 | 3 | 20.6 | 4 |
| max | 38.6 | 8 | 30.5 | 9 | 33.3 | 1 |
| mean | 32.0 | 1–10 | 27.6 | 1–10 | 27.9 | 1–10 |
| SD | 4.5 | - | 1.9 | - | 4.1 | - |
| Phosphorus | | | | | | |
| min | 2.2 | 5 | 2.4 | 6 | 1.9 | 4 and 6 |
| max | 6.8 | 1 | 4.7 | 1 | 3.4 | 1 |
| mean | 4.1 | 1–10 | 3.4 | 1–10 | 2.5 | 1–10 |
| SD | 1.3 | - | 0.7 | - | 0.6 | - |
| Potassium | | | | | | |
| min | 19.5 | 9 | 17.4 | 3 | 8.5 | 8 |
| max | 29.2 | 4 | 24.6 | 4 | 20.4 | 1 |
| mean | 23.9 | 1–10 | 21.6 | 1–10 | 16.8 | 1–10 |
| SD | 4.3 | - | 3.2 | - | 3.3 | - |
| Calcium | | | | | | |
| min | 5.4 | 7 | 0.64 | 5 | 2.24 | 10 |
| max | 11.4 | 5 | 1.18 | 3 | 0.97 | 4 |
| mean | 7.7 | 1–10 | 1.00 | 1–10 | 1.48 | 1–10 |
| SD | 2.0 | - | 0.19 | - | 0.42 | - |
| Magnesium | | | | | | |
| min | 1.5 | 2 | 2.0 | 6 and 7 | 1.4 | 7 |
| max | 3.0 | 8 | 3.2 | 3 | 3.1 | 10 |
| mean | 2.1 | 1–10 | 2.4 | 1–10 | 2.3 | 1–10 |
| SD | 0.5 | - | 0.4 | - | 0.6 | - |

The accumulation of macroelements during the growing season, per kg ha$^{-1}$, was in the following ranges: N from 162 to 366, K from 120 to 250, Ca from 50 to 100, P from 20 to 40, Mg from 10 to 20 (Figures 6a,b, 7 and 8). Despite the long-term use of galega, the results obtained are comparable to those from a shorter period of use [26,34]. The cyclical phosphorus fertilization (50 kg ha$^{-1}$ P) used in the present study covered the amount of this element removed with dry matter yields. However, the annual potassium fertilization (120 kg ha$^{-1}$ K) covered only the lower limit removed by harvested galega.

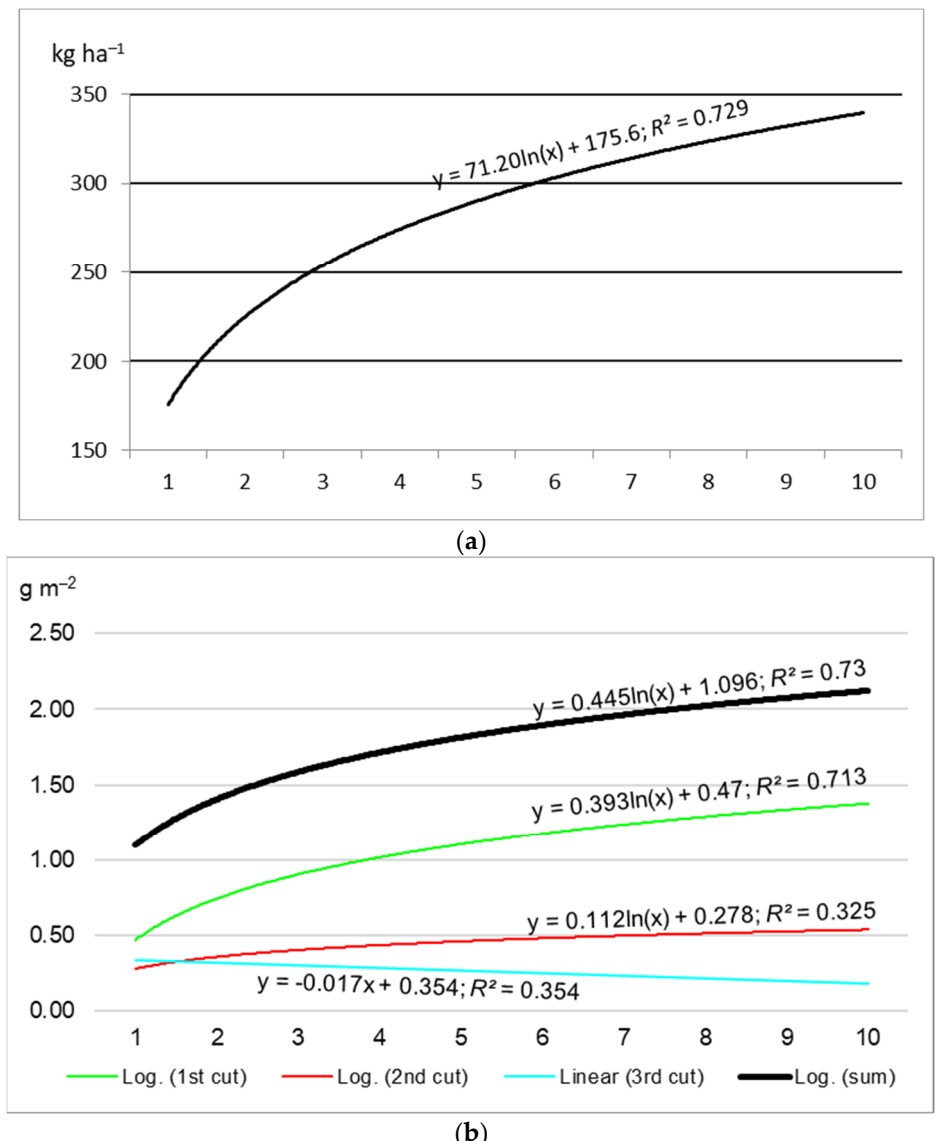

**Figure 6.** (**a**) Accumulation of total nitrogen in the total-season yield as a function of production year. (**b**) Protein yield in the total-season yield and in cuts as a function of production year.

Protein yields ranged from 1012 to 2287 kg ha$^{-1}$, with a mean value of 1777 kg ha$^{-1}$ (Figure 6b), and depended mainly on the first cut. According to Møller et al. [9], the share of protein in the yield of the first cut ranged from 46 to 70%, and in our long-term period from 44 to 71%.

Some studies on galega have focused on comparing the level and quality of yield with other perennials. The yields of alfalfa and total protein were usually higher than those of galega [9,13,24], but galega yields were higher than other less common species of perennial legumes [13]. In terms of qualitative characteristics, galega in some development stages and harvest dates was equal to alfalfa [9,27,35]. However, the advantage of fodder galega is, above all, its persistence, which significantly reduces the costs and risk of frequent establishment. It should also be added that although galega plants may be damaged by various insect species [36], apart from occasional cases of weevils feeding, no other pests or symptoms of diseases were found on galega during the course of this research, which probably contributed to the stability of the galega yields.

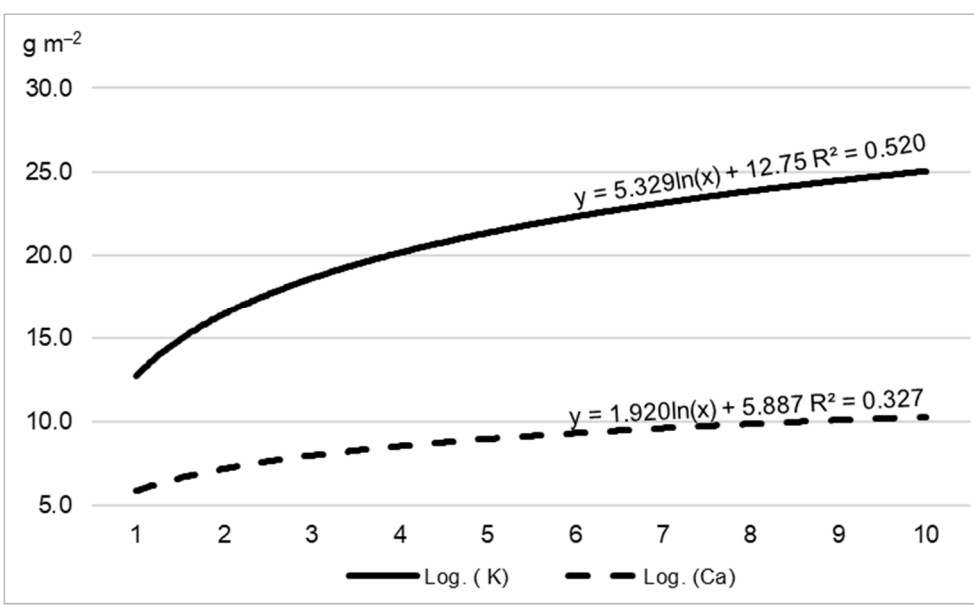

**Figure 7.** Uptake of potassium and calcium in the total-season yield as a function of production year.

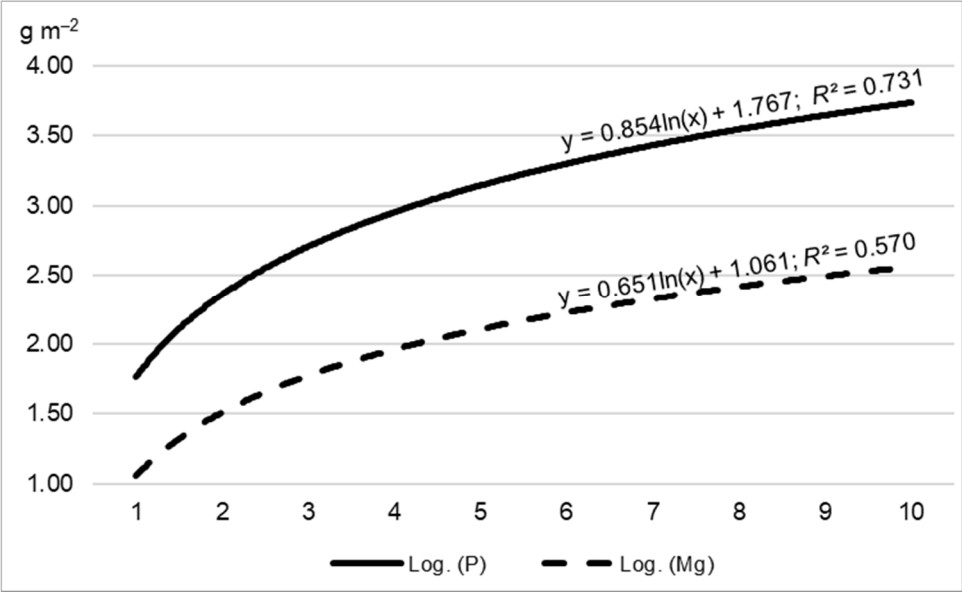

**Figure 8.** Uptake of phosphorus and magnesium in the total-season yield as a function of production year.

## 4. Conclusions

The results of many years of research on fodder galega cultivated on light soil, in a region with periodic shortages of rainfall, demonstrate that it is possible to maintain a vigorous stand for at least 10 years, with dry matter yields of 7–11 Mg ha$^{-1}$ and protein yields of 1–2.3 Mg ha$^{-1}$. However, it is crucial to maintain a constant density of shoots per 1 m$^2$ and to apply a systematic mineral fertilizer at a level that replaces the amount of nutrients removed with harvested forage. Due to the positive relationship between the vigor of plant development in the establishment year and yield in the first production year, it is justified to undertake breeding efforts aimed at improving seedling vigor and overall plant growth and development in the sowing year. Fodder galega warrants greater use in European agriculture because of its exceptional persistence and stable yield over at least 10 years; moreover, it has no need for plant protection treatments. This crop has potential

to be a reliable source of energy and protein in livestock rations as well as provide biomass for non-feed purposes.

**Author Contributions:** Conceptualization, S.I.; methodology, S.I.; performing experiments, S.I.; resources, S.I.; data curation, J.A.; software, S.I. and K.S.; writing—original draft preparation, J.A. and K.S.; writing—review and editing, J.A. and S.I.; visualization, S.I. All authors have read and agreed to the published version of the manuscript.

**Funding:** Bydgoszcz University of Science and Technology, Department of Agronomy, grant PB 610/P06/96/11.

**Data Availability Statement:** The data presented in this study are available on request from the corresponding author.

**Acknowledgments:** Jerzy Sypniewski of the holy memory for inspiring research on fodder galega in Poland.

**Conflicts of Interest:** The authors declare no conflict of interest.

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
