# Peer review of "Fodder Galega—Persistence as a Special Asset in Sustainable Agriculture"

_agronomy, doi:10.3390/agronomy13102587_

Round 1

Reviewer 1 Report

Dear colleagues!

The article submitted for review is devoted to the topical issue of studying the species Galega orientalis Lam. The crop is a valuable fodder for animals and a honey bee. The main advantage is high productivity, resistance to mowing, longevity.

G. orientalis is a promising, but still not widely spread fodder crop, which has a great future with proper cultivation and breeding work.

I consider the topic to be original, as the authors presented a generalisation of data on the results of six field experiments with G. orientalis, which were conducted in the late 1990s and early 2000s.

The results of these 10- to 20-year studies have not been previously presented to the scientific community. Therefore, I believe that they remain relevant and should be presented to the scientific community.

Unfortunately, I cannot assess the correctness of the English of the authors of the article, as I am not a native speaker of this language.

General recommendation: the article should be published. However, there are individual comments that are recommended to be addressed before the article is accepted for publication in the journal.

 The title of the article (line 3), abstract (lines 9-28) and keywords correspond to the content and results of the presented research (lines 29-30).

The aim of the work submitted for review was to evaluate the yield dynamics of G. orientalis taking into account some indicators of fodder value and accumulation of mineral components during long-term use on light soils in temperate climate with periodic rainfall deficit.

 1. Introduction (lines 32 -97).

1 The literature review is quite complete and gives an idea of the importance of the crop, problems of its cultivation. Modern literary sources are well enough represented - 26 % of references to works of the last 5 years (2018-2023).

No excessive self-citation or plagiarism was detected.

It can be advised to include in the review articles devoted to the issues of genetic research of galega:

Zolotarev, V. N. et al. “Marker-Trait Association for Breeding Fodder Galega (Galega Orientalis Lam.).” Russian agricultural sciences. (2022): n. pag. Web https://doi.org/10.3103/S1068367422040152

ÖSTERMAN, J. et al. “Galega Orientalis Is More Diverse Than Galega Officinalis in Caucasus—whole‐genome AFLP Analysis and Phylogenetics of Symbiosis‐related Genes.” Molecular ecology. (2011): n. pag. Web.. DOI : https://doi.org/10.1111/j.1365-294X.2011.05291.x

 It may be recommended to cite references to works that discuss the role of G. orientalis in the protection of fallow and contaminated soils, since the authors do not support their statement (lines 57-58) with references.

For example, see the works:

   Kaksonen, A.H. et al. “Rhizosphere Effect of Galega Orientalis in Oil-Contaminated Soil.” Soil biology & biochemistry. (2006): n. pag. Web.  Mikkonen, Anu et al. “Contaminant and Plant-Derived Changes in Soil Chemical and Microbiological Indicators During Fuel Oil Rhizoremediation with Galega Orientalis.” Geoderma. (2011): n. pag. Web. : https://doi.org/10.1016/j.geoderma.2010.10.001

  Chernyavskikh, Vladimir et al. “Invasive Activity of Galega Orientalis Lam. in the Presence of Deposits in the Southwestern Part of the Central Russian Upland.” International Journal of Environmental Studies. (2022): n. pag. Web.: https://doi.org/10.1080/00207233.2021.1987047

The purpose of the study (lines 98 -100) is clearly stated and consistent with the research conducted.

 2. Materials and methods (lines 101-154 )

In general, the section contains sufficiently detailed information on the conducted research and has a logical structure. Standard methods of soil and plant studies are listed. Statistical methods of research are characterised. All methods are reproducible. In the experiments we studied the variety Gale.

There are some questions and wishes on the section.

Questions: what is the area of experimental plots? Were these experiments to evaluate the technology of growing galega for fodder? Or was the purpose of these experiments different? Was there a control variant in the experiments?

 Recommendations: Move the reference to Table 1 to line 107 - at the end of the sentence "...typical of the cultivation of legumes.".

 3. Results (lines 155-290)

In general, the section contains sufficiently detailed information on the results of the work and has a logical structure. The obtained results are compared with the results of other authors.

1.I recommend to name the section Results and Discussion, as the authors conduct simultaneously with the description of the results of their discussions.

2. It is necessary to explain more precisely why the authors in the text compare their results with the results of 2016 and 2017, which were obtained at the same site:

«……………..However, there was a linear 186

decrease in yields in the third cut. It is worth noting, however, that in the same field in 187

2016, the yield of the first, second and third cuts of galega was 317, 349 and 385 g m2, 188

respectively, and in 2017 only two cuts were harvested—161 and 191 g m2 [25]. …… 189»

 At the same time, in section 2. the authors indicated (lines 128-131) that the longest experiment lasted from 1988 to 2008 and its results were partially described in the article [25]:

«……………..One experiment estab- 128

lished in 1988 was fully used and evaluated for 10 years, then extended for another 10 129

years. From this extended time, only selected results were collected and used. These re- 130

sults have been published in other publications [17,25] and therefore only referred to here 131

in the Discussion.………………..132»

Or did the experiment last longer - until 2017? Or was there another experiment that lasted through 2016 and 2017? Clarification is needed.

The authors have provided the necessary amount of tabular material and figures regarding the results of the research. The tables and figures fully illustrate the results.

 4. Conclusions (lines 291-300)

I recommend expanding the conclusions.

To make them more interesting for readers - to get away from general phrases and recommendations that are not related to the research conducted. In particular, nowhere in the paper (neither in the introduction nor in the discussion) was it about breeding research. But the authors write:

"Due to the great importance of the state of plant development- 297

ment in the establishment year for yields in the first productive year, it is justified to un- 298

dertake breeding work aimed at obtaining new cultivars characterised by greater vigor 299

and a faster rate of plant growth and development in the establishment year. 300"

 If the authors are going to make breeding recommendations in the conclusions, they should expand the introduction and add a discussion of scientific papers on the breeding of galega. Also expand section 3 and in the discussion consider the productivity of different varieties (according to literature data), etc.

This will enhance the readers' interest in the material presented.

 In general, the article is a completed scientific work. I recommend it for printing after editorial revisions!

Reviewer 2 Report

The manuscript does not precisely provide information about methodology and materials (cultivar or landraces of Galega) used for establishing six-long term  experiments with Galega.  The impact of meteorological factors on yield,  yield components and forage quality parameters of galega  is not shown during six long-term period. The manuscript lacks information about the material (cultivar or landrace of Galega) that was used in the experiment

English needs to be revised and improved

Round 2

Reviewer 2 Report

The corrected version of the manuscript meets all the necessary criteria for publication in the journal Agronomy